# Ornaments are equally informative in male and female birds

Sergio Nolazco [1] ✉, Kaspar Delhey [2] ✉, Shinichi Nakagawa [3] &
Anne Peters[1] ✉

Female ornaments are often reduced, male-like traits. Although these were long perceived as non-functional, it is now broadly accepted that female ornaments can be adaptive. However, it is unclear whether this is as common in females as it is in males, and whether ornaments fulfil similar signalling roles. Here, we apply a bivariate meta-analysis to a large dataset of ornaments in mutually ornamented birds. As expected, female ornament expression tends to be reduced compared to males. However, ornaments are equally strongly associated with indicators of condition and aspects of reproductive success in both sexes, regardless of the degree of sexual dimorphism. Thus, we show here in a paired comparison within-and-across species, that ornaments in birds provide similar information in both sexes: more ornamented individuals are in better condition and achieve higher reproductive success. Although limited by their correlational nature, these outcomes imply that female ornaments could widely function in a similar manner as male ornaments.

Ornaments are generally less elaborate in females than in males[1,2]. One explanation for this disparity is that female ornamentation evolved as a nonadaptive by-product of selection on male ornaments (i.e., indirect selection via cross-sex genetic correlations[1,3,4]). However, ornaments are generally assumed to be costly, and while some studies suggest that female ornaments can be non-functional[5–7], a growing number support their adaptive evolution via direct selection[8–12]. Nonetheless, it is still unclear how widespread an adaptive signalling function of female ornaments is and, crucially, how it compares to male ornaments.

Across species, the strength of sexual selection towards one sex is usually associated with the level of sexual dimorphism, with the sex exhibiting the most exaggerated or showy traits being under stronger sexual selection—typically the male[13]. Extending this argument and considering the ubiquity of cross-sex genetic correlations[4], leads to the prediction that the less elaborate ornaments of females are less likely to function in mate choice or mate competition, that is, less likely to be under sexual selection. For instance, this may occur in some dimorphic species in which ornaments play a sexually selected role in males (e.g., courtship) but that have no apparent function in females[6,7]. In such

cases, reduced expression in females could be the result of natural selection forces selecting against ornamentation to reduce potential associated costs (e.g., production or maintenance costs, or an increased predation risk).

Conversely, the tendency for reduced ornamentation in females may be the result of lower optimal expression driven by stronger trade-offs with reproduction in this sex. Whereas males often increase reproductive success by allocating more resources toward ornaments for attracting and monopolising mates[2,14], females mostly increase fitness through increased fecundity or by investing directly in offspring[15]. A larger conflict with investments in reproduction in females may be overcome by producing less elaborate ornaments. These may still provide fitness advantages not only through sexual selection (monopolising superior quality males) but also social selection (monopolising limited assets other than mates such as territory, nesting site, food, or social status)[8,10]. Indeed, the amount of investment in ornamentation does not appear to be necessarily constrained by cross-sex genetic correlations, but rather determined by sex differences in the balance between sexual/social, and natural selection[16]. Therefore, the prevalence of female ornaments to be under direct

[1]School of Biological Sciences, Monash University, 25 Rainforest Walk, Clayton, VIC 3800, Australia. [2]Max Planck Institute for Ornithology, Seewiesen, Germany. [3]Evolution & Ecology Research Centre and School of Biological, Earth and Environmental Sciences, University of New South Wales, Sydney, NSW 2052, Australia. ✉e-mail: sergio.nolazco@monash.edu; kaspardelhey@gmail.com; anne.peters@monash.edu

(sexual or social) selection, should be reflected in the average strength of their association with fitness and how this differs from males.

Trade-offs with ornament elaboration can result from costs associated with ornament production, maintenance, and/or display, with more elaborate ornaments invoking greater costs. As a result, ornament exaggeration is generally assumed to be condition-dependent. More elaborate ornaments have been predicted, and empirically demonstrated, to show greater condition-dependence in some species[17,18]. However, the generality of heightened condition-dependence has been theoretically challenged. A model by Johnstone et al.[19] shows that an increase in cost with trait exaggeration do not necessarily result in a differential of marginal costs between low- and high-condition individuals. Indeed, there are many empirical inconsistencies across taxa[20–22], suggesting that greater ornament elaboration is not generally associated with enhanced condition-dependence. However, quantitative tests at a broad scale are lacking. Understanding how patterns of condition-dependence vary in relation to ornament elaboration within and between sexes is important because condition-dependence is a central feature of evolutionary theories explaining the prevalence and variability of ornaments[14,23].

In this work, we investigated whether visual ornamentation in female and male birds provides similar information on body condition (hereafter, condition) and fitness in species in which both sexes are ornamented. We applied a phylogenetically controlled bivariate meta-analytic approach to quantify the strength and direction of associations between the degree of ornament elaboration and indicators of condition and fitness simultaneously in both sexes. Further, we tested whether these associations vary between the sexes and with the degree of ornament sexual dimorphism. While meta-analyses assessing the link between ornamentation and quality for females exist[12,24–26], a crucial quantitative comparison between males and females both within-and-across species has not been attempted. So far, a recent meta-analysis in birds has shown that colourful ornaments are, on average, honest indicators of female quality[12]. Other meta-analyses in birds (carotenoid-based ornaments)[25] and in vertebrates and invertebrates (structural-based ornament colouration)[26], incorporated male data and reached this same conclusion with no significant sex differences. However, samples were markedly male-biased and sex differences were not based on homologous ornaments within species, which is likely to affect the precision of estimates and strength of inference.

Here, we show that ornaments provide similar information content in terms of condition and fitness in female and male birds. More ornamented individuals show better condition and achieve higher reproductive success independently of sex differences in elaboration, suggesting similar potential among sexes to function as honest signals under direct selection.

## Results
### General results
We obtained 981 effect sizes ($n_{female} = 510$, $n_{male} = 471$), representing 64 species, extracted from 150 studies published between 1987 and 2019 (Fig. 1). Ornaments encompassed plumage ($n = 862$), bill ($n = 61$), eye ($n = 4$), bare skin parts such as feet, gular skin, orbital ring, wattle, comb, and gape ($n = 52$), and combinations of these traits ($n = 2$). The most common ornament types were carotenoid-based ($n = 285$), followed by melanin ($n = 188$), structural ($n = 148$), morphological ($n = 129$), unpigmented ($n = 118$), and others ($n = 113$; see Methods for details).

### Ornamentation in relation to condition and fitness combined
As a first step, we fitted a global meta-analytic model which estimates sex-specific mean effects for the relationship between the degree of ornament elaboration and all condition and fitness parameters combined. Mean effects were positive for both females and males, and

their 95% credible intervals did not overlap zero (females: $Zr = 0.19$ and 95% CI = 0.12–0.28, males: $Zr = 0.22$ and 95% CI = 0.12–0.34; model 1 in Fig. 2, Supplementary Table 1.1). Males tended to have slightly larger effects than females, but the difference was not statistically significant (female $Zr$–male $Zr$: −0.03, 95% CI = −0.15 to 0.08). Effects from correlational or experimental studies did not significantly differ (924 vs 57 effect sizes, respectively; $Zr_{study\ type} = 0.02$, 95% CI = −0.08 to 0.11, Supplementary Table 2) and, therefore, we did not consider this moderator any further.

The lack of sex differences was mostly maintained when analysing the data by ornament type (Supplementary Table 3.2). All types of traits showed statistically significant positive mean effect sizes with the exception of structural colours (model 2 in Fig. 2, Supplementary Table 3.2).

### Ornamentation in relation to condition and fitness separated
Positive effects in both sexes were larger for condition parameters than for fitness parameters, as evident from Model 3 that included a factor separating effects associated with condition or fitness parameters in interaction with sex (difference condition–fitness: $\Delta Zr$ female = 0.08, 95% CI = 0.02–0.14; $\Delta Zr$ male = 0.10, 95% CI = 0.03–0.16; model 3 in Fig. 2, Supplementary Table 1.3). The effects for condition parameters alone were positive (females: $Zr = 0.23$ and 95% CI = 0.15–0.30; males: $Zr = 0.26$ and 95% CI = 0.15–0.38). Males tended to have slightly stronger effects than females, but the difference was not statistically significant (female $Zr$–male $Zr = −0.03$, 95% CI = −0.16 to 0.08; model 3 in Fig. 2, Supplementary Table 1.3). The effects for fitness parameters alone were also positive, albeit weaker (females: $Zr = 0.14$ and 95% CI = 0.07 to 0.23, males: $Zr = 0.16$ and 95% CI = 0.05 to 0.28), with no statistically significant difference between sexes (female $Zr$–male $Zr = −0.02$, 95% CI = −0.14 to 0.10; model 3 in Fig. 2, Supplementary Table 1.3).

To determine whether different condition or fitness parameters show different associations with ornamentation, we analysed effects for specific condition and fitness parameters separately. All specific condition parameters showed positive effects, and no difference between sexes was statistically significant (Supplementary Table 3.4). Only the association between ornamentation and parasites was non-significant for males, but this can be due to the small number of effect sizes for this specific condition parameter ($n = 9$) and large associated errors (model 4 in Fig. 2, Supplementary Tables 1.4 and 3.4). For fitness parameters, we found clear positive effects for reproductive success, offspring quality or condition, and timing of breeding, while effects for parental quality and survival were not significant with 95% credible intervals substantially overlapping zero for both sexes. No differences between sexes were found for any of these effects (model 5 in Fig. 2, Supplementary Tables 1.5 and 3.5).

### Effects of sexual dimorphism
To investigate whether the degree of sexual dimorphism affects the strength of the association between ornament elaboration and condition or fitness, we tested whether effects corresponding to more sexually dimorphic ornaments show more marked differences between the sexes than effects for ornaments that do not differ much between females and males. We predicted that, as male-biased sexual dimorphism increases, male effects will become larger and more positive, while female effects will become weaker. This should translate into a statistically significant interaction between sex and sexual dimorphism.

We obtained information on ornament sexual dimorphism for 47 mutually ornamented species, and this data comprised 438 effect sizes (paired data only). As expected, sexual dimorphism tended to be male-biased (Cohen's $d$ median = 0.61 and range = −1.96–11.13, $n = 219$; positive sign indicates more ornamented males), that is, males tend to have more elaborate ornaments than females, however, mean sex

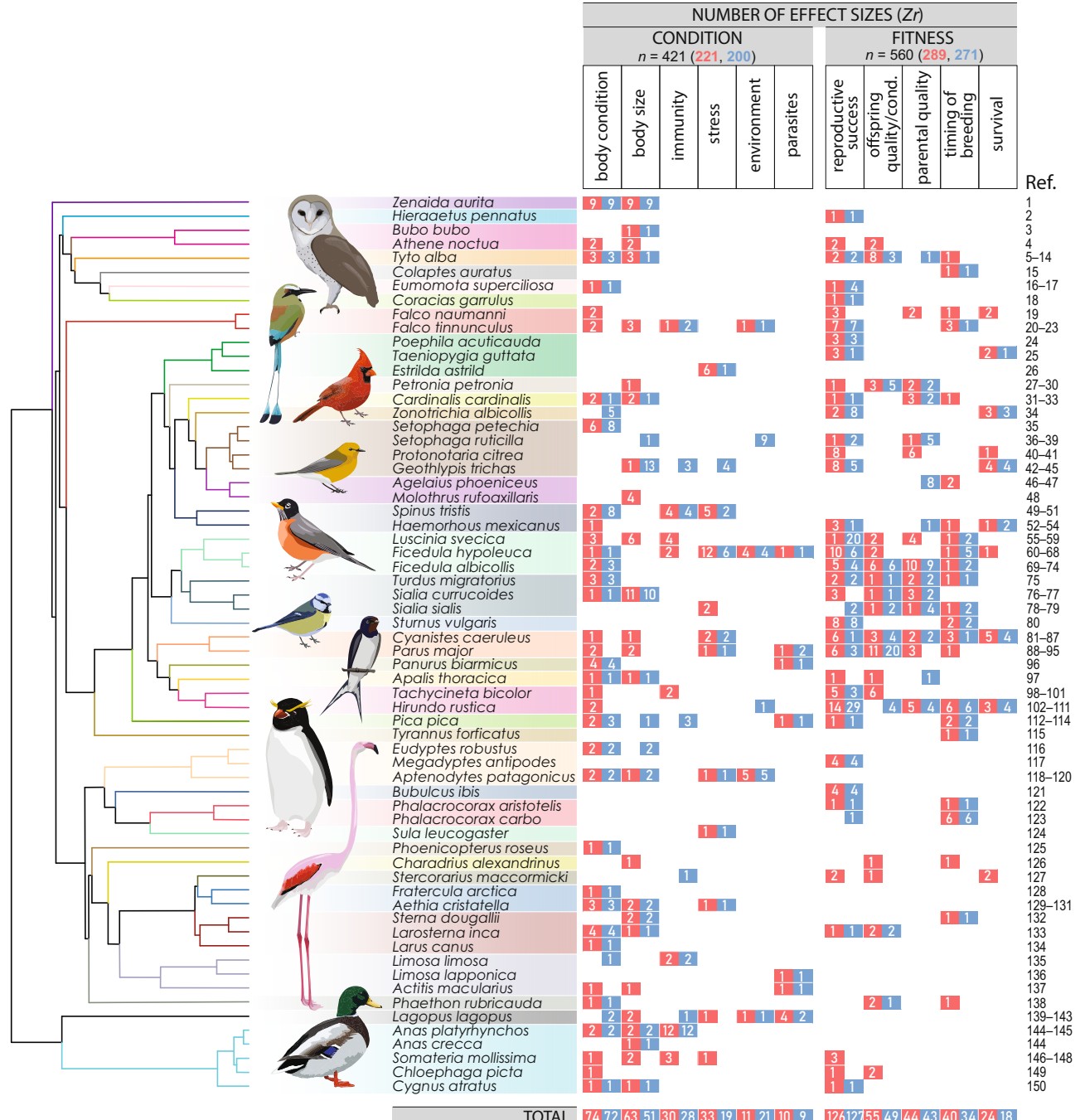

**Fig. 1 | Graphical summary of the phylogenetic distribution of mutually ornamented bird species included in our study.** Shown are the number of effect sizes for associations between degree of ornamentation and parameters of body condition and fitness per species and sex (female: red, male: blue). Taxonomic classification and nomenclature follow the Birds of the World online database[76]. Illustrations of bird species (by S. Nolazco) highlighting the taxonomic and ornament diversity of the compiled data. Colours of phylogenetic tree branches and shadows on species names represent taxonomic families (from top to bottom: Columbidae, Accipitridae, Strigidae, Tytonidae, Picidae, Momotidae, Coraciidae, Falconidae, Estrildidae, Passeridae, Cardinalidae, Passerellidae, Parulidae, Icteridae, Fringillidae, Muscicapidae, Turdidae, Sturnidae, Paridae, Panuridae, Cisticolidae, Hirundinidae, Corvidae, Tyrannidae, Spheniscidae, Ardeidae, Phalacrocoracidae, Sulidae, Phoenicopteridae, Charadriidae, Stercorariidae, Alcidae, Laridae, Scolopacidae, Phaethontidae, Phasianidae, Anatidae). Ref. 1–150 listed in Supplementary References. Source data are provided as a Source Data file.

differences were not statistically significant (Cohen's $d$ meta-analytic mean = 0.69 and 95% CI = −0.32 to 1.72). For condition parameters we initially found the expected effect of sexual dimorphism (i.e., heightened condition-dependence towards male-biased traits in males and vice versa for females), but the 95% CI nearly encompasses zero ($\beta_{\text{sex:dimorphism}}$ = 0.04 and 95% CI = 0.001 to 0.09, $n$ = 214; Supplementary Table 1.8). This outcome was driven by one highly dimorphic ornament (Willow Ptarmigan *Lagopus lagopus* comb size;

Supplementary Fig. 1). Excluding the four effects related to this ornament from the analysis reduced the effect of sexual dimorphism, which became non-significant ($\beta_{\text{sex:dimorphism}}$ = 0.03 and 95% CI = −0.05 to 0.10, $n$ = 210, Supplementary Table 1.6). For fitness parameters we found no significant effect of sexual dimorphism ($\beta_{\text{sex:dimorphism}}$ = −0.02 and 95% CI = −0.09 to 0.05, $n$ = 224; no extreme values were identified, Supplementary Table 1.7). Thus, overall variation in ornament sexual dimorphism did not affect the magnitude of the

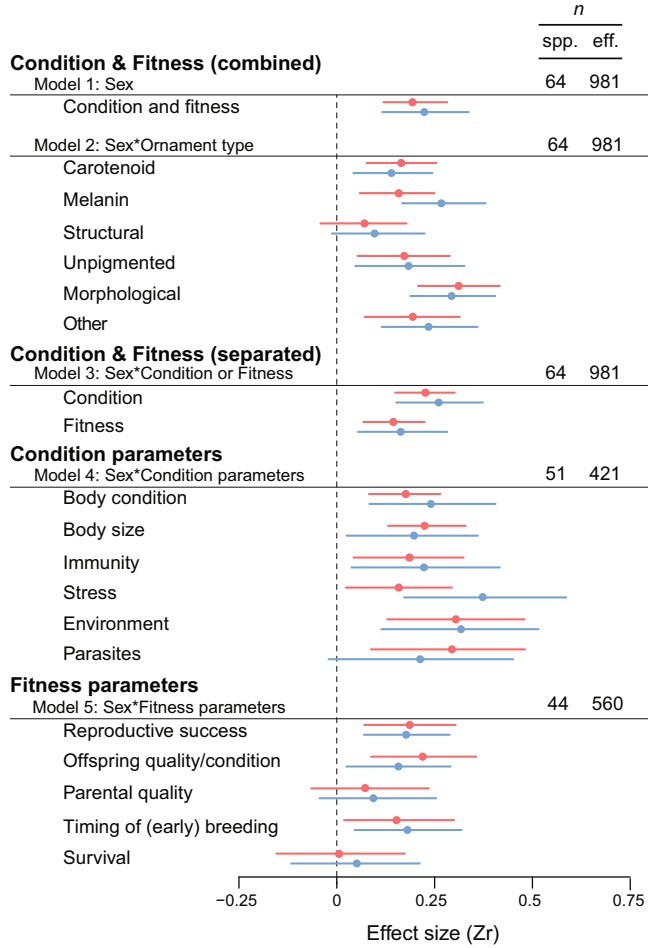

**Fig. 2 | Summary of meta-analytic models.** Shown are mean effect sizes (*Zr*) along with 95% credible intervals for each sex (female: red, male: blue). Effect sizes refer to associations between ornament elaboration and indicators of body condition and fitness investigated in populations of bird species in which both sexes are ornamented. The first model represents the global meta-analytic mean in which data on condition and fitness were pooled; the second model classified the data by ornament type; the third model classified the data in two categories (associations with condition or fitness); and the last two models classified the data into subcategories of specific indicators of condition and fitness, respectively. Number of species (spp.) and effect sizes (eff.) per model are provided. Note that the sign of the effect sizes for parasites and stress parameters were reversed, because increases in these parameters are consistent with lower condition. The same is true for timing of breeding because those individuals reproducing earlier in the breeding season, generally achieve higher reproductive success. Meta-analytic effect sizes, 95% CI, and sample sizes are provided in Supplementary Table 3. Source data are provided as a Source Data file.

association between ornament elaboration and overall condition or fitness (Fig. 3).

## Random effects and heterogeneity
Random effects, phylogeny (range of variances across models: 0.05–0.21) and species ID (0.02–0.15) had only minor effects, although in general phylogenetic effects seemed to be more marked in males than in females (Supplementary Table 4). Covariation between female and male effects tended to be stronger for fitness than for condition parameters (expressed as correlation coefficients, *r*) but in general had very broad credible intervals that overlapped zero (Supplementary Table 4). Heterogeneity (computed for the model with sex and publication year; model 1 in Fig. 2) was overall very high for females ($I^2 = 0.85$, 95%CI = 0.82–0.87) and males (0.83, 0.80–0.87).

Heterogeneity for the phylogeny component was 0.07 ($1.88 \times 10^{-10}$–0.17) for females and 0.16 ($4.39 \times 10^{-6}$–0.31) for males, and for species ID it was 0.04 ($1.50 \times 10^{-10}$–0.12) for females and 0.03 ($2.42 \times 10^{-10}$–0.11) for males.

## Publication bias
Overall, publication bias did not seem to be particularly marked for either sex. Based on exploratory analyses of funnel plots, we found slight asymmetries (i.e., seemingly minimal publication bias; Supplementary Fig. 2), also supported by Egger's tests revealing significant publication bias. Potential publication bias by trim-and-fill analysis only identified from 0 to 2 missing data points in funnel plots across datasets, all of which were negative and corresponding to relatively mid-to-low powered effect sizes found in female studies. Although these results were contradictory with Egger's tests only suggesting (negative) publication bias for male studies, adjusting for missing samples resulted in minimal mean effect sizes displacements that did not affect the conclusions (Supplementary Table 5). We did find evidence for time-lag bias ($\beta_{publication\ year} = -0.0061$ and 95% CI = −0.0089 to −0.0025; model 1, Supplementary Table 1.1), indicating that studies with larger or significant effects were published earlier than those with smaller or non-significant effects.

## Discussion
We found clear positive associations between variation in ornament elaboration and indicators of condition and fitness in males and females across bird species in which both sexes are ornamented (with low to medium effect sizes, following ref. 27). Surprisingly, none of these correlations differed significantly between sexes. Instead, our results suggest that male and female ornaments have equivalent potential to act as honest signals. Honest signalling theory proposes that ornaments provide fitness benefits to the bearer through improved reproductive success because ornamentation is a reliable indicator of individual quality[14,23,28]. Both these preconditions appear to be similarly met in males and females, irrespective of the degree of sexual dimorphism in ornamentation and preponderance of more elaborate ornamentation in males.

### Ornamentation correlates with quality and reproductive success
All mean associations between ornament elaboration and condition were clearly positive. We covered a wide range of parameters that are known to reflect condition-dependence. Associations between ornaments and these indicators of condition can arise through many pathways, alone or in combination. For instance, as a result of ornamentation relying on large energetic investments for their production or maintenance[29]; the use of rare and valuable compounds (e.g., antioxidants); the requirement of high levels of hormones with costly pleiotropic effects such as testosterone[30,31]; or intrinsic links with physiological processes that depend on condition even when ornamentation is not costly per se[32,33]. The display of ornaments can also be linked to condition due to survival costs imposed by increased predation risk[34] or social costs during agonistic interactions with conspecifics[35,36]. Our results suggest that all commonly investigated indicators of condition related to these proposed pathways to condition-dependence are linked to ornamentation. This agrees with the widely held view that ornament expression is condition-dependent.

In contrast to general condition-dependence of ornaments, we did not always find clear associations between ornament elaboration and fitness. We did find overall positive associations for indicators that are very closely related to breeding success in both sexes. These were those that quantify the actual production of offspring (reproductive success), the offspring's chances to survive and reproduce (offspring

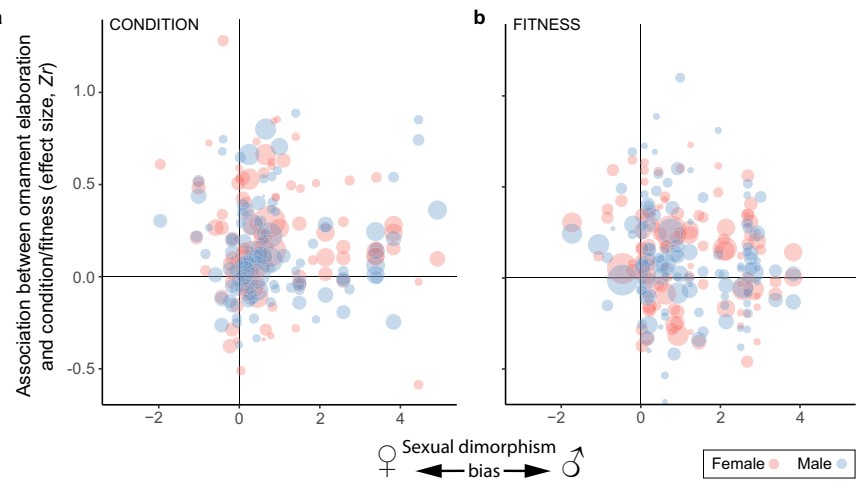

**Fig. 3 | Sexual dimorphism does not affect condition-dependence of ornaments or their association with fitness for females (red) or males (blue).** Shown are the effect sizes of ornament elaboration and (**a**) body condition and (**b**) fitness. The size of the bubbles is proportional to an inverse measure of the standard errors, thus the larger the bubble the greater the precision of the effect size. Four highly sexually dimorphic observations (outliers) were removed from the LHS plot (see Supplementary Fig. 1). Details of the models in Methods and Supplementary Tables 1.6 and 1.7). Source data are provided as a Source Data file.

quality), and early breeding (timing of breeding). In contrast, associations between ornament elaboration and parental quality and survival, were positive but non-significant with 95%CI effect sizes substantially overlapping zero for both sexes. The weak non-significant correlation with parental quality might occur because individuals can allocate differently to offspring provisioning depending on their partner's contribution or attractiveness[37]. Thus, partners that are relatively more ornamented than other individuals of their same sex may contribute less to offspring care at the expense of their mates. This might not be rare considering biparental care is generally common among mutually ornamented species[38]. Likewise, there might not be a universal association with survival because ornamentation may impose mortality costs[39] or could go undetected because most studies in vertebrates quantify survival in short periods of time rather than throughout lifetime. Above all, our outcomes indicate that in both sexes, fitness in terms of reproductive output is generally larger for those individuals investing relatively more in ornamentation within both sexes.

Our research confirmed patterns identified in a recent female-restricted quantitative review in which associations between pigment- and structural-based colour traits and condition and reproductive performance were assessed[12]. Interestingly, despite larger sample sizes in our study ($N_{condition}$ = 421 vs 46 effect sizes, $N_{fitness}$ = 560 vs 42; species number = 64 vs 47), outcomes for females were very similar, with both studies identifying equivalent meta-analytic mean association between ornamental traits and condition (ref. [12]: $Zr$ = 0.16, 95% CI = 0.10–0.23; this study: 0.23, 0.15–0.30) and near-identical mean for fitness-related parameters (0.15, 0.03–0.27; 0.14, 0.07–0.23; respectively). Although there were some minor differences in the strength of associations between individual fitness and condition parameters (most likely due to broader design and larger sample size of our study), overall, the evidence clearly shows that females showing more elaborate ornamentation are also in better condition and more reproductively successful.

Analyses separated by type of ornament indicate that, in both females and males, ornaments generally reflect individual quality, independent of their nature. For carotenoid-based ornamentation, our findings are consistent with previous meta-analyses showing positive associations between these traits and indicators of condition or fitness[12,25]. For melanin-based ornamentation, previous meta-analyses provided mixed results[12,24,40], but it is likely that our larger sample sizes may have increased the sensitivity of the analysis resulting in clear positive mean effect sizes. A common assumption is that melanin-based traits might not act as honest signals because melanogenesis is considered to be under genetic control rather than depending on nutritional sources like dietary pigments (e.g., carotenoids)[41]. However, this is currently under debate since some physiological steps during the biosynthesis and deposition of melanin can be condition-dependent[40,42]. For unpigmented ornaments our findings suggest that these traits may equally well act as honest indicators of quality as pigmented traits, which is not unexpected considering that white feathers are more vulnerable to bacterial degradation[43,44]. Morphological ornaments also showed clear links with quality, which is also not surprising considering that these traits are often expressed as anatomical extensions that are generally assumed to be costly and under direct selection[45]. The only exception to evidence for links to condition/fitness was the lack of a significant mean effect size for structural-based ornaments, despite the overall positive trend. This is different from previous meta-analyses which showed significant links of structural ornaments and individual quality[12,26]. Thus, our study together with previous studies, suggest that all these types of ornaments have the potential to broadly act as indicators of individual quality across species and sexes.

## No significant sex differences

Although we studied mutually ornamented species, ornament elaboration tended to be greater in males, which is a common pattern across bird species[14,46]. However, we did not find any statistical evidence suggestive of sex differences in the strength of the association between ornament elaboration and any measure of condition or fitness. These results were similar when considering different types of ornaments separately, and independent of the differences in ornament elaboration between sexes, since there was no significant effect of variation in sexual dimorphism for any of these associations. The lack of sex differences in condition-dependence either alone or in interaction with sexual dimorphism contrasts with the expectation that more elaborate ornaments carry larger costs that lead to increased condition-dependence[17].

Greater elaboration should entail higher costs associated with the disproportional deposition or production of pigments, or the development of complex or oversized structural traits. However, our analyses suggest that potentially greater energetic costs do not necessarily increase the relative sensitivity to condition towards the more ornamented sex. This is in accordance with theoretical models

showing that larger energetic costs that may come with further trait elaboration do not necessarily entail enhanced condition-dependence[19], and may also explain why empirical studies have provided mixed results[17,20–22]. Moreover, the apparent generality of condition-dependence for both sexes suggests that costs that are not directly involved in the production and maintenance of ornaments, such as social costs, might be more common than appreciated, in males as well as in females[36]. Combined with the lack of sex differences in the links between ornament elaboration and fitness, either alone or in interaction with sexual dimorphism, our results indicate that female and male ornaments are equally effective in capturing variance in condition and reproductive success among individuals.

The question remains, if male and female ornaments are equally good indicators of individual quality irrespective of sexual dimorphism, why are males more ornamented than females? When considering this, it should be kept in mind that our sample contained mutually ornamented species. By necessity, it lacks the more extreme versions of sexual dichromatism and polygamous mating systems where sex differences in information content may be more evident (but comparisons between the sexes would be impossible as females lack the ornamental trait). Nevertheless, our sample of species included substantial variation in sexual dimorphism, and yet the effect of this variable is weak at best (albeit positive as expected). A possible explanation is that the information content of a signal is but one aspect of optimal signal design. Indeed, signal design has two main components: a strategic design component, which constitutes the information content of the signal, and tactical design component or efficacy which deals with how the information content is conveyed to the receiver[47]. To function, signals need to be perceived and hence selection for improved efficacy may lead to more exaggerated male ornaments[45,48], if selection on signal efficacy is stronger and less restricted on males than on females. For instance, male signals may be expressed in more elaborate forms than in females in order to be detected and discriminated by receivers. The extent of exaggeration and variability in sexual dimorphism might ultimately be limited by sex- and species-specific balances between sexual/social selection and other natural selection forces (e.g., against showiness/predation). Currently, there is even less knowledge on the tactical component of signal design in females than on the strategic component, and this should be the focus of future studies if we are to fully understand how female ornaments function.

## Limitations of the study

Although our results are clear, considerable heterogeneity and the correlational nature of the dataset call for some caution. By focusing our meta-analytic bivariate approach on bird species in which both sexes are ornamented, we were able to draw conclusions from samples obtained for each sex from the same populations, investigating the same ornament parameters, and being analysed using similar methods. Although our estimates are thus based on paired data, we still obtained moderate to high between-sample heterogeneity. This is likely because we achieved large sample sizes by including a wide range of birds, study designs and parameters that do differ among studies conducted in different species. Such heterogeneity can lead to detecting apparent publication bias when no bias exists[49]. Thus, we acknowledge that our outcomes for publication bias (funnel plot asymmetries and Egger's tests) can result from causes other than publication bias. However, our results suggest that even if publication bias exist, it is relatively small across the dataset and unlikely to alter the main conclusions, which is reassuring. Another inherent limitation is that correlations between ornamentation and condition and fitness are not definitive evidence of direct selection causing the evolution of ornaments. In addition, in most cases an ornamental function, while reasonable, has not been experimentally verified, and it is possible that some ornamental traits may not be favoured by sexual or social

selection and have evolved in response to other selective forces (e.g., signalling to predators, mimicry)[50,51]. To what extent such ornaments can constitute condition-dependent traits is unclear and this may introduce some noise in our analyses weakening our inference. However, it is unlikely that such effects will obscure or create difference between the sexes and hence bias conclusions.

These limitations notwithstanding, our study does suggest that direct selection rather than cross-sex genetic correlations is a plausible explanation for the presence and maintenance of ornaments in females as in males. Overall, our synthesis of empirical research agrees with growing direct evidence in many species across taxa on the functionality and adaptive evolution of female ornaments[11,15], as well as comparative studies suggesting that these traits can evolve independently of changes in males[2,52].

## Concluding remarks

Our study shows that ornaments generally provide similar information content in females and males across bird species in which both sexes are ornamented. Our synthesis of the available information suggests that ornaments in females have equal potential to be adaptive by acting as honest signals as in males, regardless of their reduced elaboration. The lack of an effect of sexual dimorphism on the strength of associations between ornamentation and condition and fitness, suggests that greater condition-dependence and stronger selection acting on male ornaments is unlikely to be the rule despite their greater elaboration. Rather, adaptive evolution might be favouring ornamentation via honest signalling in both sexes, but with lower optimal expression in females.

# Methods

We referenced and followed the guidelines PRISMA[53] (Preferred Reporting Items for Systematic Reviews and Meta-Analyses) and PRISMA-EcoEvo[54] for writing all sections of this study (a filled PRISMA-EcoEvo checklist is provided as Supplementary Data 1).

## Literature search

We conducted a comprehensive literature search in the Web of Science (all citation databases) on 12 June 2019. We searched for publications containing the broad terms 'female ornament*' OR 'mutual* ornament*' in the title, abstract or keywords. During this search, we identified five highly cited and relevant articles on female ornamentation[2,8–10,55]. To expand our search, we also added all publications citing these articles according to the Web of Science, as well as all references cited by those five articles. The combined search strategies resulted in 849 unique references.

## Criteria for study inclusion

We went through a systematic step-by-step process of screening eligibility of publications for inclusion in the study (PRISMA flow chart in Supplementary Fig. 3). Briefly, we only included peer-reviewed articles testing associations between ornaments and condition or fitness, in bird species in which both sexes were ornamented. Only morphological ornaments that were visually recognisable and identified as such by the authors of the original studies were included, excluding traits that did not appear decorative such as body size and weapons. In all cases, ornaments were similar in structure and location between sexes. We note that in most cases ornamental function of these traits is assumed and experimental evidence is not available. Thus, we broadly defined ornaments as any phenotypic traits that look like decorations rather than having an apparent naturally selected function. Such traits seem to be beyond the natural selection optimum for functions such as crypsis, thermoregulation, or locomotion, and can be under direct (social or sexual) selection or the consequence of cross-sex genetic correlations. We excluded publications if only male traits were

investigated (i.e., no conspecific female traits were studied in this or another publication). For any female-only studies, we used the cited reference list to identify the corresponding information on conspecific males, adding 25 additional publications. Our criteria for including these studies was that the data originated from the same populations. We also identified whether studies were correlational or experimental since the later may provide stronger evidence and larger effect sizes due to manipulation. We classified as experimental those studies involving manipulations related to condition (diet, reproductive effort, wing clipping, or cross-fostering to isolate environmental from genetic effects) or ornament expression (size, colour).

All studies from which data was not extractable were excluded (e.g., studies in which insufficient data was reported such as Akaike Information Criterion, AIC, ranges of values or *P*-values alone, studies with pooled information from both sexes or incorporating sex as a covariate, or those incorporating interaction effects for ornamental traits). We used RAYYAN[56] software to manage the compiled references and to screen titles and abstracts.

### Data extraction
We obtained data in four ways: i) directly from the results section, ii) from figures using WebPlotDigitizer[57] software (version 4.5) to extract data, iii) from datasets provided in published Supplementary Materials, or iv) by contacting the authors. When several values for the same association between an ornament parameter and condition or fitness were reported, we preferentially used values stemming from linear models over simple correlations, because they often controlled for the effects of potential confounding variables. In those cases, we extracted effects from models controlling for the largest number of confounding effects. We transformed all extracted data ($F$-values, $\chi^2$, $R^2$, Mann-Whitney $U$, $t$-values, $z$-values, correlation coefficients, and comparisons between means) into correlation coefficients $r$ and then into standardised Fisher's $Zr$ effect sizes and their standard errors for comparisons between studies (following equations in ref. 58–60). When data from studies were tested by year or age class, we calculated effect sizes separately for each of these. In all cases positive $Zr$ values indicate that higher levels of ornamentation are associated with better indicators of individual quality. Thus, in some cases it was necessary to adjust the sign of the effect to reflect this, for example, studies assessing breeding timing, since in general earlier breeders achieve higher reproductive success[61].

All effects representing associations between ornamentation and aspects of individual quality were split into two broad categories: those representing associations with indicators of body condition and those representing correlations with fitness parameters. Condition parameters fell into six categories: (1) body condition: mainly measurements of body mass adjusted by structural body size and others associated with the physical condition of individuals; (2) body size: structural size (measurements of tarsus, wing, beak, keel, and tail alone or in combination) and mass; (3) immunity: indicators of constitutive immunity, immune challenges and responses; (4) stress: indicators of baseline physiological stress, stress challenges, and capacity to cope with oxidative stress; (5) environment: climatic conditions and resources; and (6) parasites: incidence and abundance of parasites. Body condition and body size are often direct indicators of nutritional status and energy balance, and immunity, parasitism, stress, and environmental quality are widely used in experimental and correlational studies to define the overall condition of individuals (reviewed in ref. 62–64).

Fitness parameters included not only estimates of reproductive success, survival and offspring quality, but also parental quality (because parental investment has been hypothesised to vary as a function of ornamentation[37]) and timing of breeding. Hence these factors were classified into five categories: (1) reproductive success: mating success and offspring production; (2) offspring quality or

condition: measurements of egg quality, offspring body condition, immunity, parasites, and other indicators of physical condition; (3) parental quality: provisioning during incubation, and offspring feeding and defence; (4) timing of breeding: measured directly or as arrival time to breeding grounds; and (5) survival.

We classified ornamental traits into six categories: (1) carotenoid-based colouration: yellow, orange, or red coloured ornaments; (2) melanin-based colouration: black, grey (eumelanin), or brown (pheo-melanin); (3) structural-based colouration: iridescent and non-iridescent; (4) unpigmented: white patches; (5) morphological: morphology of ornamental appendages (e.g., comb, wattle, tail, plumes), and (6) others: to cases in which the operational variables were a combination of two or more ornament categories or rare pigments (e.g., spheniscin in penguins). All data collection, screening, and condition and fitness parameters classification were done by a single researcher (S. Nolazco).

We extracted data on sexual dimorphism for the ornament parameters investigated from the same publications when available, or directly from the authors. We extracted information from tests comparing male and female homologous ornament parameters when available. Otherwise, we performed Welch's $t$-tests after extracting data from figures or by using the tsum.test function in the R package BSDA[65] when mean values per sex, sample sizes and measures of data dispersion were reported. We also transformed the data on sexual dimorphism into effect sizes (male in relation to female), but instead of $Zr$, we calculate Cohen's $d$ effect sizes since these are used to indicate standardised differences between means.

### Statistical analyses
We analysed the data using a bivariate meta-analytic approach with female and male effect sizes ($Zr$) as the two dependent variables to account for within-study dependencies (i.e., correlations between each pair of effect estimates). Female and male effects corresponding to the same ornament and the same condition or fitness variable were paired ($N = 470$ effects). Thus, each paired observation was generally obtained from the same study, excepting two observations that came from two studies, but conducted for the same species, region, and by the same researchers using similar methods (Supplementary References 78,79). In other cases ($N = 511$ effects), the data were only available for males or females (unpaired data). Since missing values in dependent variables are allowed within our analytic framework (see below) all effects were included, even if unpaired. However, since missing values for uncertainty estimates ($SE^2$) are not permitted, we assigned the uncertainty for the available effect to cases when only male or female effects were available. This procedure allows missing data augmentation, which results in more precise parameter estimates and constitutes a particular strength of bivariate meta-analyses[66].

We used Markov Chain Monte Carlo (MCMC) general linear mixed-effects models as implemented by function MCMCglmm in R statistical software (version 3.6.3) using the package MCMCglmm[67], which incorporated as random effects the factors study and species identities to account for multiple effects derived from the same species or study. However, we only incorporated species ID because in most cases there was only one study per species, leading to both random effects explaining roughly the same amount of variance. To control for phylogenetic relatedness, its effects were estimated by means of a phylogenetic correlation matrix. To account for phylogenetic uncertainty, we downloaded 50 phylogenetic trees (Ericson backbone) for our subset of bird species from https://birdtree.org[68] and incorporated this uncertainty into posterior distributions. The number of phylogenetic trees used was defined based on the most efficient quantity required to accurately account for phylogenetic uncertainty, as recommended by a simulation study demonstrating that 50 trees were enough to account for phylogenetic uncertainty[69].

We used uninformative parameter expanded priors for the random effects variances and covariances ($V$ = diag(2), nu = 2, alpha.mu=c(0,0), alpha.$V$ = diag(2)*625), inverse Wishart priors for the residuals ($V$ = diag(2), nu = 2) and normal distributions centred on zero with large variances as priors for the fixed effects. Models were run for 11000 iterations, with a thin of 100 and a burn-in of 1000, resulting in posterior samples of 5000. By including both the non-phylogenetic and phylogenetic species-level variance in the models we ensure to obtain approximately unbiased mean estimates and variance components[70].

We ran bivariate models with two response variables (male and female effects, specified as trait in the MCMCglmm models), and in all models we estimated separate effects for both sexes. First, we ran the intercept-only model without moderators to estimate overall meta-analytic means for females and males. Then, we included a set of moderators: a factor discriminating between condition or fitness parameters, and separately within each category, variation between different specific fitness or condition traits. We also tested for differences between ornament types (carotenoid-, melanin-, structural-based colouration, unpigmented, morphological, and others) including it as a moderator in a separate model. We also tested for the effects of sexual dimorphism and study design (correlational/experimental). Our models are unlikely to be underpowered by the inclusion of these moderators since all models have relatively large sample sizes. We report mean effects point estimates along with 95% credible intervals (CI) obtained from posterior distributions using the function HPDinterval.

Publication bias was assessed using meta-analytic residuals separately for each sex using funnel plots, Egger's regression tests and the trim-and-fill method using the package metafor[71]. Funnel plots are a graphical representation of effect sizes against a measure of study precision (e.g., variance, standard error). Visual inspection of asymmetries on funnel plots serve as a sign of publication bias, formally tested using Egger's test[72]. The trim-and-fill method was applied to identify and correct for funnel plot asymmetries. This method inputs missing samples (i.e., effect sizes) until symmetry of funnel plot is reached[73]. We compared significance of mean effect size change before and after applying the trim-and-fill method using a $Z$-test. Relatively low changes in mean effect sizes with small number of missing values can be interpreted as minor potential publication bias. Another type of bias that is generally overlooked in meta-analyses is known as the time-lag bias[74]. Time-lag bias occurs when those studies with larger or significant effects are published quicker than those with smaller or non-significant effects, which translates into a decline over time. We tested and controlled for this bias by including publication year (mean centred to zero) as a moderator following ref. 74. We did not find evidence for an interaction between publication year and sex ($\beta_{sex:publication\ year}$ = 0.002 and 95% CI = −0.004 to 0.01), so we only incorporated additive effects to control for time-lag bias. We also computed heterogeneity ($I^2$), a statistic that describes variation in outcomes across studies, following ref. 75.

### Reporting summary

Further information on research design is available in the Nature Research Reporting Summary linked to this article.

## Data availability

A complete raw data file, along with calculated effect sizes (Supplementary Data 2), the prepared dataset (Supplementary Data 3) and the subset of phylogenetic trees (Supplementary Data 4) used to run the analyses, and a list of all references from which the data were extracted (Supplementary Information) are also provided. Phylogenetic trees can be downloaded directly from https://birdtree.org and species nomenclature used in this paper is available from https://birdsoftheworld.org (accessed on April 2020). Source data are provided with this paper.

## Code availability

R code for reproducing the results in this paper is available as Supplementary Code 1.

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

## Acknowledgements
S.No. is grateful to Monash University for funding his PhD studies with the Faculty of Science Dean's International Postgraduate Research Scholarship (DIPRS), the Graduate Research Completion Award (GRCA) and the Postgraduate Publications Award (PPA). A.P. acknowledges funding from the Australian Research Council (FT110100505; DP180100058; DP210100328) and the School of Biological Sciences, Monash University. Special thanks to researchers that provided additional data from their published studies that were required to calculate effect sizes for sexual dimorphism or associations between ornamentation and condition and fitness parameters, or provided clarification of the ornament parameters analysed: Alexandre Roulin, Andrea S. Grunst, Amélie Dreiss, Bettina Almasi, David López-Idiáquez, Juan Moreno, Matthew W. Reudink, Mike W. Butler, Miklós Laczi, Pierre-Paul Bitton, Pierre Legagneux, Raivo Mänd, Roxana Torres, Simon C. Griffith, Susan L. Balenger, Tuul Sepp, Vallo Tilgar, Vicente García-Navas.

## Author contributions
S.No., K.D. and A.P. designed the study, S.No. carried out the systematic literature review, extracted the data and ran the analysis with K.D. S.No., K.D. and A.P. wrote the manuscript. S.Na. made key contributions to the developing of the R code to build the bivariate models and contributed to the writing.

## Competing interests
The authors declare no competing interests.
