## [Peer Review File · Nature Communications]

Ornaments are equally informative in male and female birdsReviewers' Comments:

Reviewer #1:

Remarks to the Author:

This meta-analysis compares the relationship between visual ornament elaboration and body condition or fitness components in males and females across mutually ornamented bird species. Unexpectedly, similar condition-dependence of ornament expression is found in both males and females, on average. This is an interesting and well-presented study that informs questions of long-standing interest in sexual selection regarding explanations for exaggerated ornaments, and sexual dimorphism in ornamentation.

My comments and questions are mostly minor, and are given by line numbers below. A more general comment is around the implications of your findings for honest signalling theory. You focus on the potential for ornaments in both sexes to honestly signal condition, regardless of the extent of ornament elaboration. Wouldn't this also reduce the power of the honest signalling hypothesis to explain ornaments? I might be thinking about this the wrong way, but your results seem to suggest that while ornaments are on average honest signals of condition (as non-ornamental traits might also be, e.g. general plumage condition, body fat) this is not why they are maintained or exaggerated. I guess I am thinking from the direction why is optimal expression higher in males (rather than lower in females)? Why grow a peacock tail if a peahen tail does the job? (metaphorical example only ;)

L32 How do you define 'ornament' in your study? Are ornaments necessarily under sexual selection in both sexes, or in at least one sex? It seems this could be quite subjective – for example, if a male ornament is elongated tail feathers, what determines whether female tails are also ornaments or simply the baseline expected tail length (compared to which male tails appear 'exaggerated')?

L46 To me, 'evolutionary disappearance' implies that females once displayed ornaments as exaggerated as males. Further, I'm not sure that traits expressed in females due to cross-sex genetic correlations are in the process of disappearing – rather they are expressed above their female optimum by the genetic correlation (instead of by trade-offs). Can you clarify what you mean here?

L120 Did you have a method to 'randomly' select one effect size where several were presented in a study?

L118-122 Re-reading this sentence, that does not sound like random selection but rather a set order of preference for certain types of effects over others. This sentence needs to be edited for clarity.

L158-160 Was the missing sex more often females or males? Was there any relationship between the extent of dimorphism and the availability of ornament measures in both sexes? I also wondered what the effect of so many missing pairs might be on the bivariate model, and whether substituting the error for the available sex in so many cases affects the model estimates (or just allows the model syntax to run)?

L332 Perhaps it is worth mentioning that funnel plot asymmetry and significant Eggers' tests may result from other causes than publication bias – see e.g. Peters et al. 2008, *Journal of the Royal Statistical Society A*, 173:575-591 and references within.

- L440 I see you are aware of these issues – it then seems a bit odd to present the funnel plots and Eggers' tests simply as evidence for publication bias in the Results (e.g. "Table S5: Publication bias does occur..."). I think these caveats to the interpretation of tests for publication bias should be mentioned in the Results as well as the Discussion.

L360-363 I would have liked to see this justification for the aspects of condition you chose to include presented earlier, in the Methods section. And similarly, explanation of the fitness components included.

L403 This is not really an adage – maybe ‘axiom’ is more what you’re getting at here? It would be good to find a reference putting forward this idea.

Reviewer #2:

Remarks to the Author:

This is an interesting comparative analysis of female “ornaments” in birds. The authors use data from 64 species and 150 studies to examine the strength of the relationship between traits (plumage, bill and eye) color or size and various measures of condition (eg, body mass) or “fitness” (e.g., fledging success, survival). They found that these traits provide similar information about condition or fitness in both sexes, although the traits are generally reduced in size or color in females compared to males. This is an important result, but a recent meta-analysis in *Biology Letters* (Hernandez et al. 2021) has also reported a positive relationship between female color traits and body condition or reproductive performance. The study here goes beyond the Hernandez analysis by comparing the correlation in males and females, but the results still seem limited and confirmatory, rather than revealing something new. Hence, I would like to see a more detailed rationale for publishing in this journal rather than a more specialized evolution journal. I also have two other concerns about the paper (described in more detail below). One is the absence of any discussion of natural selection on female ornaments. The focus here is almost entirely on sexual selection. Second, the authors use the term “ornament” for traits that differ in color or patch size, but in most of these studies we do not know if the traits are sexually selected through mate choice or competition. It is possible that many traits are not sexually selected and, hence, not ornaments in the typical meaning. The fact that the authors find relationships despite this potential noise in the system is interesting, but they also need to acknowledge the limitation of the data. Otherwise I think the analyses and writing are good.

1) p. 2. Introduction. “with the sex exhibiting the most exaggerated traits being under stronger selection – typically the male 15–18.” I would change to “stronger sexual selection”. The authors should be careful not to mix natural and sexual selection, because there is natural selection on plumage in both sexes (Dunn et al. 2015. *Science Advances* 10.1126/sciadv.1400155). Similarly, in the next sentence: “leads to the prediction that the less elaborate ornaments of females are less likely to be under direct selection, that is, less likely to have an adaptive function.” This should be reworded to “under sexual selection” and “less likely to function in mate choice” Or something along those lines. I am not exactly sure what to change this to, but a reduced female ornament could have an adaptive function in camouflage or territoriality, just not be important for sexual selection.

2) p. 2 Introduction. “reduced ornamentation in females can be an evolutionary snapshot of a process of ornament disappearance.” Actually, if you look at evolutionary transitions in dimorphism (or color of each sex individually), there are equal rates of change for males and females (Dunn et al. 2015. *Sci Adv.*), so the authors need to think about this more broadly in terms of what causes changes in color; it is not just sexual selection as this introduction is implying.

3) p. 3 Introduction. “Consequently, females may express less elaborate ornaments to reduce conflicts with investments in reproduction, without being necessarily constrained by cross-sex genetic correlations 23–27.” Again, the authors are focusing on “reproduction” (and sexual selection?) and apparently forgetting about natural selection. Females also experience natural selection on their ornaments.

In general, I think the authors should start out with a more balanced view of female ornaments and talk about both natural and sexual selection influencing ornaments, then they can go on to talk about sexual selection. I have no problem focusing mainly on sexual selection on female ornaments, but the authors need to acknowledge natural selection and then move on.

4) p. 9-10. The models are a bit hard to understand, so it would help if the authors explicitly stated that Model 1 examines differences in effect sizes between males and females, and it used all the data; ie, correlations between ornaments and both condition and fitness. I would have done condition separately from fitness first (Model 2) and then combined them (ie, reverse model 1 and 2), since the data are so heterogeneous (eg, body mass vs nest visits), but I guess this is okay.

By the way, I could not find a "readme" file for the supplemental file titled "343921_0_source_data_149736_r32kxl.csv"
So it is unclear what all the variables mean.

5) p. 20. I would not say that the Hernandez et al. study was "much smaller" because they used 46 species compared with 64 here for female effect sizes. I suspect that the number of effect sizes in this study is much larger because the authors are using more response variables in each study. They also appear to have included more studies per species. For example, just picking a species at the top of the list, *Aethia cristatella* (Crested auklet), the authors obtained effects sizes for the relationship between female body mass and both "plumage auricular crest" and "plumage crest" from two studies, whereas Hernandez appears to have only used one study (Jones et al. 2000). By the way, this makes me wonder if the two "crests" mentioned here are really different traits. Does this effect the analysis in any way? Ie, I am wondering how different (or the same) traits are handled for the same species (within an analysis of, say, 'condition'). Are these crests treated as one or two traits?

6) After looking at the data, I see that the authors also include some variables that I would not include as "condition-dependent". Again, looking at the auklet, I see wing length, which is usually not considered condition dependent in the sense of some of these other traits that are more labile and change yearly. Thus, if you compare these two studies more closely, then I think they will be more similar in sample size etc, although I do note that there are some large differences between the effect sizes in each study for *Setophaga ruticilla* and *Phalacrocorax aristotelis*. Those should probably be double-checked, if possible.

7) p. 22. Conclusions. "Our results also suggest that maladaptation or non-functionality of female ornaments is generally unlikely or rare. Rather, adaptive evolution might be favouring ornamentation via honest signalling in both sexes, but with lower optimal expression in females."

This seems like a strange choice of words (non-functionality or maladaptation?) given that female plumage probably evolved for a lot of other reasons besides sexual selection. I think the authors are, again, not acknowledging the important role of natural selection on female "ornaments". I would also point out that we really know very little about the function of these "ornaments" and what appear to be ornaments to us (humans), could have a different (non-sexually selected) function, especially if they are just a little patch of color. This should also be mentioned under limitations of the study.

RESPONSE TO REVIEWERS

REVIEWER COMMENTS

Reviewer #1 (Remarks to the Author):

This meta-analysis compares the relationship between visual ornament elaboration and body condition or fitness components in males and females across mutually ornamented bird species. Unexpectedly, similar condition-dependence of ornament expression is found in both males and females, on average. This is an interesting and well-presented study that informs questions of long-standing interest in sexual selection regarding explanations for exaggerated ornaments, and sexual dimorphism in ornamentation.

My comments and questions are mostly minor, and are given by line numbers below. A more general comment is around the implications of your findings for honest signalling theory. You focus on the potential for ornaments in both sexes to honestly signal condition, regardless of the extent of ornament elaboration. Wouldn't this also reduce the power of the honest signalling hypothesis to explain ornaments? I might be thinking about this the wrong way, but your results seem to suggest that while ornaments are on average honest signals of condition (as non-ornamental traits might also be, e.g. general plumage condition, body fat) this is not why they are maintained or exaggerated. I guess I am thinking from the direction why is optimal expression higher in males (rather than lower in females)? Why grow a peacock tail if a peahen tail does the job? (metaphorical example only ;)

Authors' reply: This is an interesting point, and to fully answer it we probably have to also measure the absolute cost of ornaments, not only the relative difference in expression between males and females. Nevertheless, beyond their information content (the strategic component of signal design), ornaments need to be perceived and assessed for the signal to be successful. This tactical component of signal design, more commonly known as signal efficacy, may explain why on average males are more ornamented than females despite similar levels of information content. Thus, ornaments may be more elaborate in males if efficacy selection is more important and less restricted than in females. We have elaborated on this possibility in L340-353, although we stress that this efficacy aspect of signal design is even less studied than information content in females and should be the focus of future studies.

L32 How do you define 'ornament' in your study? Are ornaments necessarily under sexual selection in both sexes, or in at least one sex? It seems this could be quite subjective – for

example, if a male ornament is elongated tail feathers, what determines whether female tails are also ornaments or simply the baseline expected tail length (compared to which male tails appear 'exaggerated')?

Authors' reply: We have now added a very broad definition of what we are calling ornaments, and which fits with our compiled data (L418-421). In each case we also followed the decision of the authors of the original studies to corroborate this as they have first-hand experience with their study species (we now mention this in L414). However, we agree with the reviewer that in many cases there is no clear experimental evidence that ornaments are used as signals in either females or males. We have added this caveat to L370-376, L416-417. We note also, that if putative 'ornamental' traits in females are not exaggerated beyond the naturally selected optimum, this should bias our results towards finding a difference in effect between males and females. This was not the case.

L46 To me, 'evolutionary disappearance' implies that females once displayed ornaments as exaggerated as males. Further, I'm not sure that traits expressed in females due to cross-sex genetic correlations are in the process of disappearing – rather they are expressed above their female optimum by the genetic correlation (instead of by trade-offs). Can you clarify what you mean here?

Authors' reply: We have re-written this section to clarify (L43-49).

L120 Did you have a method to 'randomly' select one effect size where several were presented in a study?

Authors' reply: see response to next comment.

L118-122 Re-reading this sentence, that does not sound like random selection but rather a set order of preference for certain types of effects over others. This sentence needs to be edited for clarity.

Authors' reply: We have edited this sentence to clarify the process (L442-444). Selection was not random as the reviewer noticed. Thank you for spotting this issue.

L158-160 Was the missing sex more often females or males? Was there any relationship between the extent of dimorphism and the availability of ornament measures in both sexes? I also wondered what the effect of so many missing pairs might be on the bivariate model, and whether substituting the error for the available sex in so many cases affects the model estimates (or just allows the model syntax to run)?

Authors' reply: Males have more missing values than females, but the difference was not large (275 vs 236). Unfortunately, we could not test whether missingness was associated with sexual dimorphism, since we need data on males and females to obtain sexual (ornament) dimorphism. To indicate that dimorphism values were only available from studies in which both male and female traits measurements were available, we changed the sentence in L179.

We replaced uncertainty values for missing effects in one sex with those of the other sex for two reasons: First, as the reviewer says, this allowed the MCMCglmm models to run, but second, and more importantly, this procedure allows missing data augmentation by the MCMCglmm algorithm which results in more precise parameter estimates and constitutes a

particular strength of bivariate meta-analyses (Jackson et al. 2017 *Stat Methods Med Res* 26:2853–2868). We now added this clarification to the Methods (L504-506).

L332 Perhaps it is worth mentioning that funnel plot asymmetry and significant Eggers' tests may result from other causes than publication bias – see e.g. Peters et al. 2008, Journal of the Royal Statistical Society A , 173:575-591 and references within.

- L440 I see you are aware of these issues – it then seems a bit odd to present the funnel plots and Eggers' tests simply as evidence for publication bias in the Results (e.g. “Table S5: Publication bias does occur...”). I think these caveats to the interpretation of tests for publication bias should be mentioned in the Results as well as the Discussion.

Authors' reply: We have added the requested clarification (L364-367).

L360-363 I would have liked to see this justification for the aspects of condition you chose to include presented earlier, in the Methods section. And similarly, explanation of the fitness components included.

Authors' reply: We have shifted the explanation to the Methods section as requested (L463-469).

L403 This is not really an adage – maybe ‘axiom’ is more what you’re getting at here? It would be good to find a reference putting forward this idea.

Authors' reply: We have changed “adage” with “expectation” and added the requested reference (L318-319).

Reviewer #2 (Remarks to the Author):

*This is an interesting comparative analysis of female “ornaments” in birds. The authors use data from 64 species and 150 studies to examine the strength of the relationship between traits (plumage, bill and eye) color or size and various measures of condition (eg, body mass) or “fitness” (e.g., fledging success, survival). They found that these traits provide similar information about condition or fitness in both sexes, although the traits are generally reduced in size or color in females compared to males. This is an important result, but a recent meta-analysis in *Biology Letters* (Hernandez et al. 2021) has also reported a positive relationship between female color traits and body condition or reproductive performance. The study here goes beyond the Hernandez analysis by comparing the correlation in males and females, but the results still seem limited and confirmatory, rather than revealing something new. Hence, I would like to see a more detailed rationale for publishing in this journal rather than a more specialized evolution journal.*

Authors' reply: We thank the reviewer for the comments and interest. We respectfully disagree that the results are limited or confirmatory. In addition to a much larger sample size (981 compared to 88 effect sizes in Hernández et al. 2021), our study reveals entirely novel patterns and conceptually advances the field by quantitatively comparing effects in males and females in the same species and for the same traits.

We achieve this by running phylogenetically controlled bivariate meta-analyses. This has never been done before in this context, but it provides the crucial necessary contrast to determine to what extent males and females differ. Based on theory, if ornamentation is mainly selected for

in males and its expression in females is but a consequence of genetic correlation, we would have expected substantially weaker effects in females compared to males. This was not the case, effects in males and females are very similar and the difference is never statistically significant. This result is unexpected and changes our thinking about the function of ornamentation in males and females.

Moreover, we further tested for the role of sexual dimorphism by collating, for each putative ornamental trait separately, a standardised measure of dimorphism. This allowed us to test whether ornaments that are more elaborate in males than in females would show stronger links with condition and fitness in males than females. This was not the case, further extending the general results. In sum, we show for the first time, using state-of-the-art analytical approaches and a large detailed dataset, that ornaments can be equally informative of quality in males and females. Since our approach and outcomes are entirely novel, we think that our results go beyond what could be considered confirmatory. We have tried to clarify the novelty of our approach further in L22-23 and L83-94.

I also have two other concerns about the paper (described in more detail below). One is the absence of any discussion of natural selection on female ornaments. The focus here is almost entirely on sexual selection. Second, the authors use the term “ornament” for traits that differ in color or patch size, but in most of these studies we do not know if the traits are sexually selected through mate choice or competition. It is possible that many traits are not sexually selected and, hence, not ornaments in the typical meaning. The fact that the authors find relationships despite this potential noise in the system is interesting, but they also need to acknowledge the limitation of the data. Otherwise I think the analyses and writing are good.

Authors’ reply: (1) We now mention the possibility that natural selection could be behind some of the putative ornaments included and that this may constitute a source of noise in our analyses (L370-376).

(2) What is an ornament? We agree with the reviewer that for many/most traits considered here we do not have experimental evidence that they indeed are signals used by conspecifics to assess individual quality. We follow the assumptions of the original authors of each study which deemed these traits potential ornamental signals. We now clarify this in Methods (L414) and state that this is a limitation, which is probably shared by most studies on sexual selection (L370-371).

1) p. 2. Introduction. “with the sex exhibiting the most exaggerated traits being under stronger selection — typically the male^{15–18}.” I would change to “stronger sexual selection”. The authors should be careful not to mix natural and sexual selection, because there is natural selection on plumage in both sexes (Dunn et al. 2015. Science Advances 10.1126/sciadv.1400155).

Authors’ reply: Changed accordingly (L41).

Similarly, in the next sentence: “leads to the prediction that the less elaborate ornaments of females are less likely to be under direct selection, that is, less likely to have an adaptive function.”

This should be reworded to “under sexual selection” and “less likely to function in mate choice” Or something along those lines. I am not exactly sure what to change this to, but a reduced female ornament could have an adaptive function in camouflage or territoriality, just not be important for sexual selection.

Authors' reply: We have changed the sentence following the reviewer suggestions, **from:** “leads to the prediction that the less elaborate ornaments of females are less likely to be under direct selection, that is, less likely to have an adaptive function” **to:** “leads to the prediction that the less elaborate ornaments of females are less likely to function in mate choice or mate competition, that is, less likely to be under sexual selection” (L42-44).

Regarding being careful not to mix natural selection and sexual selection, we have rewritten the second and third paragraphs of the *Introduction* (L43-49, 54-61) to avoid conceptual confusions. However, we find it unlikely that the ornamental traits studied here may work as camouflage in females. Instead, natural selection may often push ornaments to become less elaborate or showy in one sex, generally in females, and this may affect the signalling functions. We also acknowledge that ornaments can play a role in territoriality (i.e., as an armament) as the reviewer states. Therefore, now we have also explicitly indicated in the third paragraph of the *Introduction* that ornaments can also evolve through social selection (L57-58). We have also made explicit that the level of expression of ornaments can be the result of sexual and social selection in balance with other natural selection forces that may act in different directions.

2)p. 2 *Introduction*. “*reduced ornamentation in females can be an evolutionary snapshot of a process of ornament disappearance.*” *Actually, if you look at evolutionary transitions in dimorphism (or color of each sex individually), there are equal rates of change for males and females (Dunn et al. 2015. Sci Adv.), so the authors need to think about this more broadly in terms of what causes changes in color; it is not just sexual selection as this introduction is implying.*

Authors' reply: We agree with the reviewer that evolutionary changes in sexual dimorphism can be caused by changes in the male, the female or both, which could be indicative of the relative balance of sexual, social and natural selection acting on each sex. We have changed this sentence and cited examples to clarify our message (L43-49).

3)p. 3 *Introduction*. “*Consequently, females may express less elaborate ornaments to reduce conflicts with investments in reproduction, without being necessarily constrained by cross-sex genetic correlations²³⁻²⁷.*” *Again, the authors are focusing on “reproduction” (and sexual selection?) and apparently forgetting about natural selection. Females also experience natural selection on their ornaments.*

Authors' reply: We now explicitly acknowledge the role of social selection and that the optimal ornamental expression is the result of the balance between sexual, social, and natural selection (L54-61).

In general, I think the authors should start out with a more balanced view of female ornaments and talk about both natural and sexual selection influencing ornaments, then they can go on to talk about sexual selection. I have no problem focusing mainly on sexual selection on female ornaments, but the authors need to acknowledge natural selection and then move on.

Authors' reply: We hope that our revisions have achieved the more balanced approach requested by the reviewer.

4)p. 9-10. *The models are a bit hard to understand, so it would help if the authors explicitly stated that Model 1 examines differences in effect sizes between males and females, and it used all the data; ie, correlations between ornaments and both condition and fitness. I would have*

done condition separately from fitness first (Model 2) and then combined them (ie, reverse model 1 and 2), since the data are so heterogeneous (eg, body mass vs nest visits), but I guess this is okay.

Authors' reply: We now state explicitly that Model 1 tests for differences between the sexes using all data (L103-105). We have followed standard procedures to present meta-analytic results, starting with the intercept only model to compute the overall means. In this case we start combining data for associations between ornamentation and condition and fitness to test for an overall mean effect. Since we used a bivariate model, we still separated these data by sex. The second stage is to test for differences between groups, in this case between those types of associations, whether it is condition or fitness and so on (subcategories within these parameters).

*By the way, I could not find a "readme" file for the supplemental file titled "343921_0_source_data_149736_r32kxl.csv"
So it is unclear what all the variables mean.*

Authors' reply: We thank the reviewer for noticing that we did not attach a "Readme" file for the Raw Data. Now we have changed the Raw Data file from .csv to excel workbook and attached a Readme within that file.

5)p. 20. I would not say that the Hernandez et al. study was "much smaller" because they used 46 species compared with 64 here for female effect sizes. I suspect that the number of effect sizes in this study is much larger because the authors are using more response variables in each study. They also appear to have included more studies per species. For example, just picking a species at the top of the list, Aethia cristatella (Crested auklet), the authors obtained effects sizes for the relationship between female body mass and both "plumage auricular crest" and "plumage crest" from two studies, whereas Hernandez appears to have only used one study (Jones et al. 2000). By the way, this makes me wonder if the two "crests" mentioned here are really different traits. Does this affect the analysis in any way? Ie, I am wondering how different (or the same) traits are handled for the same species (within an analysis of, say, 'condition'). Are these crests treated as one or two traits?

Authors' reply: In accordance with the reviewer's point of view about the relative size of Hernández et al. study compared to ours (not based on sample size but number of species), we have changed sentences in L278-279. We have also moved this sentence together with the rest of the paragraph to accommodate it according to changes done in the discussion and the addition of a new paragraph discussing new analyses on ornament types (L288-308).

Regarding the Crested auklet referenced example, we note that the effects we extracted do not refer to a "plumage auricular crest", but to the "plumage auricular plume", which is a different trait compared to the "plumage crest" (e.g. rows 698-699 in Dataset 1: Supplementary Dataset, article citation is indicated in Supplementary materials text file as [129] Jones et al. 2000). We did also extract data for these traits from other two studies (Douglas et al. 2009, Klenova et al. 2011) that, unlike Jones et al. 2000, provided enough data to calculate effect sizes for both sexes for a same trait (rows 3, 622-625 in Dataset 1: Supplementary Dataset).

6)After looking at the data, I see that the authors also include some variables that I would not include as "condition-dependent". Again, looking at the auklet, I see wing length, which is usually not considered condition dependent in the sense of some of these other traits that are more labile and change yearly. Thus, if you compare these two studies more closely, then I

think they will be more similar in sample size etc, although I do note that there are some large differences between the effect sizes in each study for *Setophaga ruticilla* and *Phalacrocorax aristotelis*. Those should probably be double-checked, if possible.

Authors' reply: We agree with the reviewer's comment that some indicators of individual condition can be more labile than others. This is why we assigned specific indicators of condition into several different categories (L456-463) and we also tested them separately (L164, Fig. 2). For the Crested auklet we also used data on wing length (extracted from Klenova et al. 2011 study; citation 131 in Supplementary Material text file) as the authors used this measure as an indicator of body size. Body size can be determined by multiple factors such as genes and diet during development, and it is usually a trait selected by mate choice or as an indicator of competitive or fighting capabilities. However, we understand that this measure can be less variable than others (e.g., mass corrected by size or experimental treatments of nutritional manipulation), but still an indicator of condition on a different temporal scale.

The reviewer also identified some differences in effect sizes between our study and Hernández et al. (2021). We thus went back to the original publications and data and carefully checked these potential discrepancies. We mostly found our estimates to be correct (with one minor error for *Setophaga ruticilla*, as indicated below).

For *Phalacrocorax aristotelis* females, our effect size is $Z_r = -0.65$ and $SE = 0.21$ (row 130, Dataset 1: Supplementary Dataset, crest size vs laying date) which is exactly the same point value and error as reported by Hernández et al. (Hernández et al. 2021: row 52 in their dataset <https://doi.org/10.5061/dryad.bg79cnpb7>), but their effect is negative rather than positive. We considered it a positive effect (because in birds earlier laying dates are generally indicative of higher individual quality and those attempts have higher reproductive success). Thus, in this case (as for all laying dates effects) we reversed the sign of the negative association between crest size and laying date before running the analyses (good_fem column in Dataset 1) as explained in (L451-453).

Hernández et al. also reported another effect size from the same source (Daunt et al. 2003) for the association between crest size and fledging success (Hernández et al. 2021: row 53 in their dataset). They provided a significantly large (negative) effect size ($Z_r = -1.40$ and $SE = 0.04$; row 53 in their dataset). We think this is a mistake since the referenced paper indicated no significant differences: "We found no significant difference in crest size between unsuccessful and successful breeders, i.e. that fledged at least one young (median and interquartile range: females unsuccessful: 3.75, 2.75–4.17 (n = 19); females successful: 3.81, 2.94–4.31 (n = 6); $U = 51.00$)". In addition, the value should be positive as successful females have a larger median value. We calculated the effect size (non-significant $Z_r = 0.08$, $SE = 0.21$) using the Mann-Whitney statistics provided in the paper. This same result was confirmed after performing a Welch t-test using the means and standard deviations calculated from the provided medians and IQ ranges. We had not included this effect size because it stemmed from a non-parametric Mann-Whitney test. However, as we since have been alerted to the fact that it is possible to extract these effects (Sidney 1956 *Nonparametric statistics for the behavioral sciences*, Krishnamoorthy 2016 *Handbook of statistical distributions with applications*) we now have included the calculated effect size for females, and also males (when available from the same papers), and went back to check all our compiled articles to see if we have overlooked other similar results. In total, we have added 14 new effect sizes increasing our number of effect sizes from 967 to 981. All analyses have been re-run but no conclusion has changed.

For *Setophaga ruticilla*, we compiled effects from four studies while Hernández et al. (2021) compiled only one (Osmond et al. 2013). The mismatched effect corresponds to the association between brightness of carotenoid patch (tail) and fledging success, where we obtained $Z_r = -0.65$, $SE = 0.29$, while Hernández et al. 2021 obtained $Z_r = 0.56$, $SE = 0.19$ (row 75 in their dataset). Thus, there are two important differences: the magnitude of the effect and its sign. The effect was obtained from the reported association indicating that “SY (second year) females with brighter tails fledged less offspring” (second paragraph in *Results* from the cited paper; Osmond et al. 2013). We suspect that Hernández et al. (2021) used the wrong sample size to calculate the effect size. We recalculated the effect size using the t -value and degrees of freedom and we obtained the value reported by Hernández et al. ($Z_r = 0.56$) when assuming that sample size is 30. This is probably because in Osmond et al. (2013) there is no explicit mention of the sample size for the association of interest, but they report sample sizes ranging from 27 to 30 for other analyses. The correct sample size should be 15 based on the model results reported in Osmond et al. 2013 ($F[2,13] = 6.29$, $P = 0.01$; $df_2 = n - 2$), which can be confirmed by checking the plot in that same paper (Fig. 2A). We also noted that we made a mistake for this effect because we used the F -value corresponding to the overall model to calculate the effect size. This is not correct since there was more than one covariate in the model. We now corrected the effect size by using the t -value reported in the cited paper ($t = -3.10$), which resulted in a Z_r value of -0.78 (row 374, Dataset 1). Unlike, Hernández et al. 2021 we decided to keep the effect size as negative since Osmond et al. (2013) found that older females (ASY; after second year) had brighter tails, and they also cited other papers indicating that higher quality males exhibit brighter tails.

Finally, we have double-checked the rest of observations in our datasets and found no further issues. We have also updated all results, tables and figures, as well as our datasets, with no changes in any of our conclusions.

7) p. 22. *Conclusions.* “Our results also suggest that maladaptation or non-functionality of female ornaments is generally unlikely or rare. Rather, adaptive evolution might be favouring ornamentation via honest signalling in both sexes, but with lower optimal expression in females.”

This seems like a strange choice of words (non-functionality or maladaptation?) given that female plumage probably evolved for a lot of other reasons besides sexual selection. I think the authors are, again, not acknowledging the important role of natural selection on female “ornaments”. I would also point out that we really know very little about the function of these “ornaments” and what appear to be ornaments to us (humans), could have a different (non-sexually selected) function, especially if they are just a little patch of color. This should also be mentioned under limitations of the study.

Authors’ reply: We now mention in the limitations that ornamental signalling functions are generally assumed and that the required experimental results are lacking (L370-371), and we have edited to text of the Conclusion section (L385-393) by removing any reference to maladaptation.

Reviewers' Comments:

Reviewer #1:

Remarks to the Author:

I enjoyed reading the revised version of this manuscript. The authors have comprehensively addressed all my comments and those of the other reviewer, and I think the manuscript explains key ideas and methods more clearly as a result of their changes. I tend to agree with the authors regarding the value of their study and its suitability for this journal – I think this unexpected lack of sex difference provides much food for thought and further investigation, and will be of interest to a broad audience.

I have no further comments for the authors.

Reviewer #2:

Remarks to the Author:

The authors have done a good job of revising the paper so it is more balanced now. I think it is suitable for publication now.